# Iteratively Prompting Multimodal LLMs to Reproduce Natural and AI-Generated Images

**Ali Naseh, Katherine Thai, Mohit Iyyer & Amir Houmansadr**
University of Massachusetts Amherst
{anaseh,kbthai,miyyer,amir}@cs.umass.edu

## Abstract

With the digital imagery landscape rapidly evolving, image stocks and AI-generated image marketplaces have become central to visual media. Traditional stock images now exist alongside innovative platforms that trade in prompts for AI-generated visuals, driven by sophisticated APIs like DALL-E 3 and Midjourney. This paper studies the possibility of employing multi-modal models with enhanced visual understanding to mimic the outputs of these platforms, introducing an original attack strategy. Our method leverages fine-tuned CLIP models, a multi-label classifier, and the descriptive capabilities of GPT-4V to create prompts that generate images similar to those available in marketplaces and from premium stock image providers, yet at a markedly lower expense. In presenting this strategy, we aim to spotlight a new class of economic and security considerations within the realm of digital imagery. Our findings, supported by both automated metrics and human assessment, reveal that comparable visual content can be produced for a fraction of the prevailing market prices ($0.23 - $0.27 per image), emphasizing the need for awareness and strategic discussions about the integrity of digital media in an increasingly AI-integrated landscape. Additionally, this approach holds promise as a tool for data augmentation, potentially enhancing machine learning models by providing varied and cost-effective training data. Our work also contributes to the field by assembling a dataset consisting of approximately 19 million prompt-image pairs generated by the popular Midjourney platform, which we plan to release publicly.[1]

## 1 Introduction

In recent years, image stocks and marketplaces have become increasingly important in the commercial and business sectors. Alongside traditional stock images, known for their high quality and compositions by expert photographers, a new trend has emerged in the form of marketplaces for AI-generated images, such as PromptBase,[2] PromptSea,[3] and Neutron Field.[4] Unlike traditional stocks where the images themselves are traded, these innovative platforms trade in the prompts that lead to the creation of AI-generated images. Advanced text-to-image APIs like DALL-E 3 (Betker et al., 2023) and Midjourney,[5] with their extraordinary ability to generate stunning visuals, are at the forefront of this trend. However, identifying the right prompts to produce such images is not a straightforward task (Cao et al., 2023; Oppenlaender, 2023), leading to the development of marketplaces where users can exchange their crafted prompts.

With the growing demand for purchasing prompts from AI-generated sources, and the continued interest in traditional stock images, a pivotal question emerges: *Can an adversary*

---

[1]https://github.com/SPIN-UMass/MidjourneyDataset
[2]https://promptbase.com
[3]https://www.promptsea.io
[4]https://neutronfield.com
[5]https://www.midjourney.com

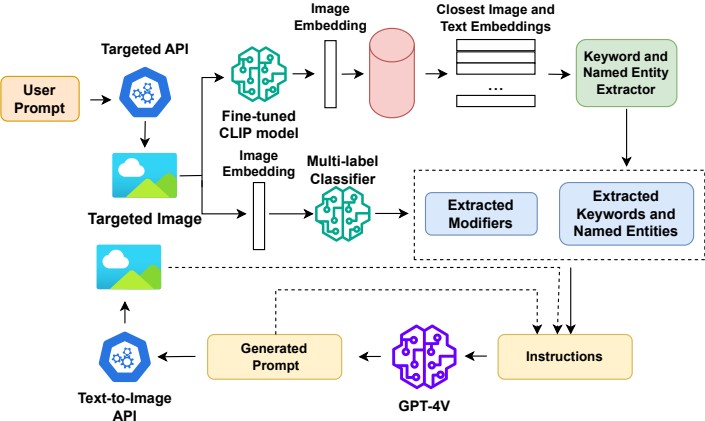

Figure 1: Overview of our attack.

*find a prompt and utilize current state-of-the-art text-to-image models to generate similar images at a lower cost?* This question becomes particularly significant when considering that some of the natural images in traditional stocks are very expensive.[6] This paper investigates this query, demonstrating how the latest multi-modal models with visual understanding capabilities can be harnessed for such attacks against these marketplaces. Our study exposes potential vulnerabilities in the digital imagery landscape and emphasizes the necessity for strategic countermeasures in response to these evolving challenges.

In this paper, we first demonstrate that if an adversary is given images generated by one of the text-to-image APIs featured in AI-generated image marketplaces, they can, by accessing one of these APIs, find a prompt to generate similar images. Additionally, we show that the same attack methodology can be applied to generate images that closely resemble natural images offered in traditional stock markets. This dual capability of the attack strategy underscores its potential impact on both AI-generated and natural image domains.

*Can we simply utilize off-the-shelf image captioning models to recover the prompts used to generate an input image?* Current image captioning models (Li et al., 2023; 2022a; Wang et al., 2022; Chen et al., 2022) often produce general descriptions that capture the broad aspects of an image but typically lack the specificity required for text-to-image API prompts like those used in Midjourney. The textual input prompts for these APIs need to be more specific and follow a particular format, usually by incorporating keywords and key phrases that significantly influence the generation (Oppenlaender, 2023). Directly using captions from standard image captioning models may not yield effective results since they often omit critical details and stylistic elements. While tools like CLIP Interrogator[7] can suggest corresponding text and keywords, they may not provide enough detail and are especially limited when describing natural images. Similarly, although multimodal models like Gemini[8] and GPT-4V provide more detailed descriptions, they might still overlook essential keywords or named entities.

To bridge the gaps presented by these tools, we introduce a three-component attack strategy: a fine-tuned CLIP model (Radford et al., 2021) on a large dataset of Midjourney generations, a multi-label classifier for related keywords and named entities, and GPT-4V for its ability to generate comprehensive prompts based on our instructions and information from the CLIP model and the classifier. We then implement a cyclic approach to refine these prompts, comparing the generated images with the original (ground-truth) image(s). The overview of our attack strategy is illustrated in Figure 1.

---

[6]https://www.gettyimages.com/detail/photo/mother-and-graduating-daughter-pose-cheek-to-cheek-royalty-free-image/1366624125?phrase=graduation+parent

[7]https://huggingface.co/spaces/fffiloni/CLIP-Interrogator-2

[8]https://deepmind.google/technologies/gemini/#introduction

Besides highlighting the potential threats to digital imagery, our approach also serves as an effective tool for data synthetic and augmentation. By generating varied visual content using different random seeds with the final prompts created by our pipeline, we can produce diverse image sets. This capability is particularly valuable considering that traditional data collection is often time-consuming and expensive. These generated image sets can enhance the training of machine learning models across various applications, providing a cost-effective alternative to real data acquisition (Tian et al., 2024).

As a significant contribution of our work, we have collected a large-scale dataset consisting of 19,137,140 generations from Midjourney's Discord server. This dataset, which pairs each prompt with its corresponding image or gridded image, aids in fine-tuning the CLIP model and training the multi-label classifier to identify related keywords and named entities. Our approach is validated through both automatic metrics and human evaluation, confirming that our attack outperforms existing baselines. Additionally, we provide a cost analysis to justify the feasibility of the attack. **Our cost estimation indicates that an attacker can generate a reasonably similar image to the targeted one at a cost of only $0.23 to $0.27**.

To summarize our contributions:

1. To the best of our knowledge, this work presents the first systematic investigation of potential attacks against images generated by two of the most popular text-to-image APIs, as well as against natural images from commercial stock collections.

2. We demonstrate that an attacker can generate images similar to those produced by AI, typically priced between $3 and $7, and natural stock images valued between $70 and $500 per image, all at a mere fraction of the cost — just a few cents.

3. A significant portion of our effort was dedicated to collecting and preprocessing a substantial dataset of Midjourney's generations. This dataset is not only pivotal to our project but also stands as a valuable resource for future research in this field.

## 2 Related Work

### 2.1 Text-to-Image Generation Models

The journey of text-to-image generation began with methods based on Generative Adversarial Networks (GANs) (Goodfellow et al., 2014). These GAN-based models paved the way for the field, focusing on synthesizing visual content from textual descriptions (Reed et al., 2016; Mansimov et al., 2015; Zhang et al., 2017; Xu et al., 2018; Zhu et al., 2019). In recent years, the emergence of diffusion models (Sohl-Dickstein et al., 2015) and large pre-trained autoregressive models has led to the introduction of many new text-to-image models (Ramesh et al., 2021; Ding et al., 2021; Cho et al., 2020). These developments introduce multimodal transformer language models, which are proficient at learning the distribution of sequences of discrete image codes from given text inputs. Parti (Yu et al., 2022) introduces a sequence-to-sequence autoregressive model treating text-to-image synthesis as a translation task, converting text descriptions into visuals. Imagen (Saharia et al., 2022) employs a large language model for quality text features and introduces an Efficient U-Net for diffusion models. Latent Diffusion Models (LDM) (Rombach et al., 2022), such as Stable Diffusion (Rombach et al., 2021), employ diffusion processes within the latent space. This approach facilitates efficient model training on systems with constrained computational resources, without compromising the fidelity and quality.

### 2.2 Prompt Stealing Attack

Prompt stealing attacks represent a domain closely related to our work. In such attacks, the objective of the attacker is typically to reconstruct the original prompt used in generating an image. However, our goal is to replicate AI-generated and natural images, where the resulting prompt does not necessarily need to match the original one precisely, word-by-word. Although this area has seen limited exploration, there have been a few metods in the context of both Large Language Models (LLMs) (Zhang & Ippolito, 2023; Sha & Zhang,

2024) and Text-to-Image models (Shen et al., 2023). Concurrently, Liu et al. (2024) attempts to reverse-engineer the textual prompt to generate a specific image for later customization, highlighting another approach within this emerging field.

## 3 Threat Model

### 3.1 Attacker's Capabilities.

In our proposed attack, we consider an attack scenario that adheres to a realistic paradigm where the attacker is granted black-box access to all text-to-image APIs. Specifically, the attacker lacks access to the underlying model and its training data; their interaction is confined to providing input and receiving the corresponding output. A key assumption in our model is that the attacker possesses inference-time access to the API and also access to the generations of one of these APIs, namely Midjourney's Discord server. This is a realistic assumption, as we have access to it. Furthermore, we extend our consideration to a more complicated scenario where the attacker gains access to an alternate API, distinct from the primary target, to facilitate the attack. Additionally, in the case of natural images, the attacker only has access to the targeted image(s) and nothing more.

### 3.2 Attacker's Objective.

The attacker's goal in this scenario is to identify a prompt that can generate images similar to a given image or set of images, whether produced by a text-to-image API or featured in one of the commercial stock image collections. This objective necessitates that the attacker has the capability to send queries to the API to execute the attack. As previously noted, our investigation also includes scenarios where the attacker utilizes an alternative API, distinct from the primary target, to carry out the attack.

### 3.3 Attacker's Target.

In our attack scenario, the main targets are two well-known text-to-image APIs: Midjourney and DALL-E 3. We aim to target the images generated by these APIs in order to find prompts that produce similar images. Additionally, for natural images, we focus on images from the Getty Images website,[9] which represents one of the popular image stock sources.

## 4 Methodology

Our approach to executing the attack centers around three primary components: keyword extraction, modifier extraction, and prompt generation. Each plays a crucial role in the attack strategy's overall effectiveness, as detailed in subsequent subsections.

**Keyword Extraction.** As previously noted, image captioning models often lack the specificity required for text-to-image API prompts, typically omitting crucial details and stylistic elements needed for APIs like Midjourney. Adding to this challenge, a key limitation of these models is their inconsistent accuracy in extracting specific keywords, special names, or named entities depicted in the images. Although they capture some details, they often lack the precision needed for completely accurate prompt replication. This combination of generalization and lack of precision in keyword and entity extraction significantly hinders their effectiveness in accurately generating prompts for text-to-image generation. An example of BLIP2's failure to detect the keywords is shown in Figure 2.

To address this challenge, we employ a substantial dataset obtained from Midjourney, consisting of a subset of 2 million pairs of prompts and corresponding upscaled images generated by Midjourney. We then fine-tune a CLIP model on this dataset. The CLIP model, initially pre-trained on 2 billion pairs of captions and images, many of which are

---

[9]https://www.gettyimages.com/

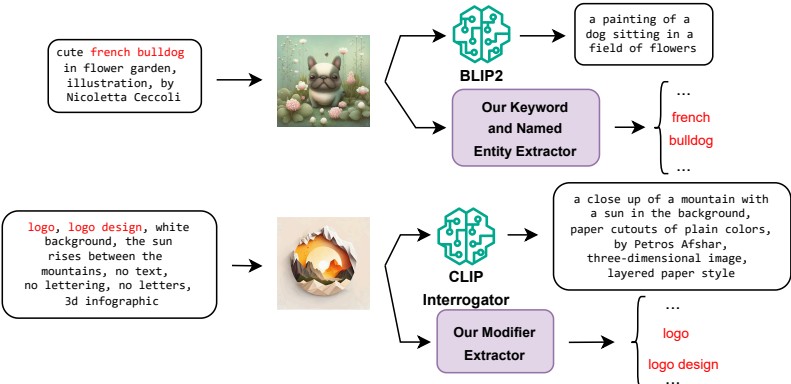

Figure 2: Examples illustrating scenarios where the baseline models, BLIP2 and CLIP Interrogator, fail to accurately extract relevant keywords and modifiers from given prompts.
.

not precisely aligned, requires fine-tuning on the Midjourney subset to better align with the nature of the prompts and images in our study.

The fine-tuned CLIP model is then utilized to generate a set of 5 million text and corresponding image embeddings. During the inference phase, we use the fine-tuned CLIP model to obtain image embeddings of the targeted image(s) and identify the closest text and image embeddings. From these, we extract a list of named entities and keywords from the most closely related prompts and the corresponding prompts of the nearest images. Finally, this information, along with other relevant data, is fed into GPT-4V.

**Modifier Extraction.** Besides keywords and named entities, incorporating adjectives, adverbs, and specific styles, referred to as modifiers, can significantly enhance the quality of generations (Oppenlaender, 2023). However, image captioning models typically are unable to detect these modifiers, failing to recognize and extract them accurately. Additionally, the CLIP Interrogator also does not consistently perform well in detecting these modifiers. An example of the CLIP Interrogator's failure to detect modifiers is illustrated in Figure 2. To overcome this, we train a Multi-Layer Perceptron (MLP) as a multi-label classifier on image embeddings. After identifying 1800 frequent modifiers from our Midjourney dataset, we train the multi-label classifier on a selected subset of around 300, 000 samples. During inference, we determine the top modifiers with the highest probability scores. These modifiers, along with other pertinent information, are then provided to GPT-4V.

**Prompt Generation.** The final step involves using a model with strong visual understanding capabilities for detailed image description. For this, we select GPT-4V, a multi-modal model known for its exceptional visual understanding. However, GPT-4V alone may not always suffice. To bridge this gap, we complement it with detailed instructions and information derived from the other two components. Our comprehensive instructions to GPT-4V, called *initial instructions*, include a detailed task description, relevant modifiers, general keywords, named entities, and an example prompt from Midjourney samples. While the fine-tuned CLIP model and the multi-label classifier provide relevant information, GPT-4V acts as an additional classifier, selecting the most appropriate modifiers, keywords, and named entities. This leads to a two-level extraction process. All the related information, along with the targeted images, is fed into GPT-4V to generate the prompt.

**Refinement of the Initial Prompt.** The initial prompt generated by GPT-4V might not always capture every detail in the image. To enhance the prompt quality, we implement a cyclical refinement process. This involves devising a new set of instructions for GPT-4V, including elements like a detailed task description, relevant modifiers, keywords, named entities, an example from Midjourney, the targeted image(s), the prompt used in the previous round of the cycle, and the corresponding generated image(s). The task description also

| Model | Top-1 Accuracy | Top-5 Accuracy | Top-10 Accuracy |
|---|---|---|---|
| Original CLIP model | 0.802 | 0.915 | 0.9395 |
| Fine-tuned on 200K samples | 0.8065 | 0.9413 | 0.967 |
| Fine-tuned on 500K samples | 0.9003 | 0.9653 | 0.9783 |
| Fine-tuned on 2M samples | **0.9167** | **0.9762** | **0.9857** |

Table 1: Comparative metrics of the original CLIP and fine-tuned models on varying dataset sizes, highlighting improvements in Top-1, Top-5, and Top-10 accuracy.

emphasizes comparing the generated image(s) with the targeted ones, refining the prompt based on differences in styles, themes, or elements. This iterative process continues over multiple rounds to progressively refine the prompt's accuracy and relevance. The initial and refining prompts are displayed in Table 6 and Table 7, respectively. The overview of the attack is presented in Figure 1.

## 5 Experimental Settings and Results

### 5.1 Midjourney Dataset

As previously mentioned, Midjourney stands as a frontier text-to-image API, known for generating high-quality and realistic images. Users interact with Midjourney's bot on their official Discord server to input prompts and generate images, making these generations accessible to anyone who joins the server. Acknowledging the importance of this API, we collect millions of samples from Midjourney's Discord channels. This substantial dataset is crucial for our attack, providing a wide array of prompts and corresponding images. Details about the dataset and its pre-processing can be found in the Appendix A.

### 5.2 Fine-Tuning the CLIP Model

**Model Configuration and Dataset Details.** For the task of fine-tuning, we select the CLIP ViT-G/14 model (Ilharco et al., 2021), which is pre-trained on a vast collection of 2 billion samples from the LAION dataset (Schuhmann et al., 2022). We pick this model variant as it shows better performance in ImageNet zero-shot classification tasks.[10] Through iterative experimentation, we optimize the hyperparameters, setting a learning rate (lr) of $1 \times 10^{-5}$, a batch size of 32, and gradient accumulation steps to 2. Resource constraints lead us to fine-tune the CLIP model on a 2 million sample subset of the Midjourney dataset. The outcomes of fine-tuning with various dataset sizes are shown in Table 1.

**Metric and Evaluation Results.** To evaluate the fine-tuned CLIP models, we use a test set of 10,000 samples, employing CLIP's image and text encoders to generate embeddings. We calculate cosine similarity for each test image to identify the closest corresponding text, measuring the model's accuracy by whether the ground-truth text is within the Top-1, Top-5, or Top-10 closest texts. For example, a Top-5 accuracy of 90% indicates that the ground-truth text ranks among the top-5 closest texts for 90% of the samples. Table 1 presents the evaluation metrics for both the original and the fine-tuned CLIP models across different dataset sizes. The fine-tuned model significantly outperforms the original for this data type, with

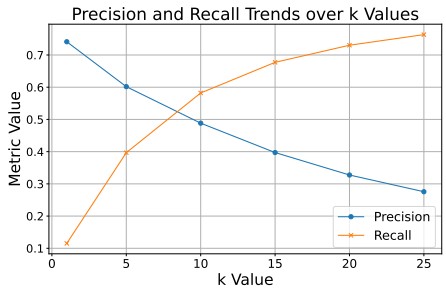

Figure 3: Overview of the multi-label classifier's performance for different values of $k$.

---

[10]https://github.com/mlfoundations/open_clip

| Setting | BLIP2 | | CLIP Interrogator | | Our Attack | |
|---|---|---|---|---|---|---|
| | CLIP-S | HE | CLIP-S | HE | CLIP-S | HE |
| Midjourney (multiple images) | 0.8477 | 3.4 | 0.8874 | 3.88 | **0.8946** | **4.36** |
| Midjourney (single image) | 0.7483 | 3.6 | 0.8437 | 4.04 | **0.8844** | **4.52** |
| DALL-E 3 (multiple images) | 0.8534 | 4.16 | 0.8569 | 3.68 | **0.8883** | **4.56** |
| DALL-E 3 (single image) | 0.7932 | 2.8 | 0.8007 | 2.84 | **0.896** | **4.08** |
| Cross-setting (multiple images) | 0.798 | 3.8 | 0.8169 | 4.04 | **0.859** | **4.4** |
| Cross-setting (single image) | 0.7911 | 3.04 | 0.8215 | 3.28 | **0.886** | **4.08** |
| Natural images (single image) | 0.7516 | 3.8 | 0.748 | 3.92 | **0.8031** | **4.36** |

Table 2: Comparison of different methods using CLIP-S (CLIP-score) and HE (Human Evaluation) metrics across various settings.

the model fine-tuned on 2 million samples showing the best performance, though the improvement over other dataset sizes is marginal.

### 5.3 Training Multi-Label Classifier

**Model Configuration and Dataset Details.** We utilize a Multi-Layer Perceptron (MLP) architecture with three hidden layers for this task. The ReLU activation function is employed, along with a learning rate of 0.001. To mitigate overfitting, we incorporate a dropout rate of 0.3. Upon analyzing the Midjourney dataset, we identify approximately 1800 distinct modifiers. From this, we select around $360,000$ samples, labeling them based on the presence of these modifiers. We allocate 10% of this dataset as a validation set to monitor and adjust model performance. Detailed information about the process of extracting these modifiers can be found in the Appendix A.5.

**Metrics and Evaluation Results.** For this task, we utilize precision and recall as the primary metrics. During inference, the top $k$ modifiers are selected based on their scores from the classifier, where precision indicates the accuracy of selected modifiers and recall assesses the proportion of correctly predicted ground-truth modifiers. Figure 3 displays precision and recall values for different $k$ thresholds, highlighting a trade-off: increasing $k$ lowers precision, reducing the proportion of correctly selected modifiers, but enhances recall, capturing more correct modifiers. Though higher recall is preferable for comprehensive modifier coverage, avoiding excessive inclusion is essential. To balance these metrics, we choose $k = 20$ for our overall attack evaluation.

### 5.4 Overall Attack Evaluation

**Experimental Settings.** We evaluate two major text-to-image APIs, Midjourney and DALL-E 3, for AI-generated images and Getty Images website for natural images. Our scenarios include both multiple images from a single prompt and single image cases for Midjourney and DALL-E 3. We also explore an adversary using a different API, with Midjourney as the target and DALL-E 3 as the alternate, across multiple and single image variations. This results in seven distinct settings, with 150 samples for setting 1, 50 samples each for settings 2-4, and 30 samples for settings 5-7.

**Model Configuration and Dataset Details.** In scenarios 1, 2, and 7, we utilize Midjourney as the text-to-image model for generating images. In scenarios 3-6, DALL-E 3 is employed for image generation, with the output size set to $1024 \times 1024$ and standard quality. Across all scenarios, GPT-4V serves as the multimodal model for visual understanding and prompt generation. For scenarios involving Midjourney-generated images, a small subset of the

| Setting | Original Image | Our Attack | CLIP Interrogator | BLIP2 |
|---|---|---|---|---|
| Midjourney (Setting 2) | | | | |
| DALL-E 3 (Setting 4) | | | | |
| Cross-setting (Setting 6) | | | | |

Figure 4: Comparison of the original image with those generated by our method, BLIP2, and CLIP Interrogator for three different settings.

Midjourney dataset, not used in CLIP model fine-tuning or multi-label classifier training, is selected. In scenarios where DALL-E 3 generates the targeted images, we utilize a subset of Midjourney prompts to generate the corresponding images, which then form the basis of our evaluation dataset. For scenarios involving natural images, we carefully select a diverse range of samples from the Getty Images website.

**Metric.** To assess the similarity between the images generated by our approach and the ground-truth images, we employ both automated metrics and human evaluation. The automated metric, termed Clip-score, involves calculating cosine similarity between the image embeddings from the original CLIP model provided by OpenAI. Additionally, for human evaluation, we select five random samples from each setting (35 samples in total) and ask five annotators to rate these images. They use a 5-point Likert scale to score the images based on perceived similarity. More details about the human evaluation process are presented in Appendix B.

| Text-to-Image API | BLIP2 | CLIP Interrogator | Our Attack |
|---|---|---|---|
| DALL-E 3 | 0.7509 | 0.7044 | **0.8284** |
| Midjourney | 0.7516 | 0.7480 | **0.8132** |

Table 3: Image similarity scores for our attack on natural images using different text-to-image APIs, compared with baseline models.

**Baselines.** Our evaluation includes BLIP2, a recent image captioning model, and CLIP Interrogator 2, as baselines. For scenarios involving a single image, we use the textual output provided by these baselines directly. In settings with multiple images, to ensure a fair comparison, we first process each image through the baselines and then select the one with the highest similarity score.

**Evaluation Results.** The results for all seven settings are presented in Table 2. Across these settings, our approach consistently outperforms the baselines. The margin of superiority is substantial in most settings, except for the first, where CLIP Interrogator shows close performance based on CLIP-score. It's observed that the effectiveness of all methods diminishes in scenarios 5 and 6, likely due to the use of a different API by the attacker. Furthermore, the results show improved performance in scenarios involving multiple images compared to those with a single image, likely because multiple images provide more comprehensive information for analysis. Some examples of generated images by our approach and other baselines are presented in Figure 4.

### 5.5 Natural Images: DALL-E 3 vs Midjourney

In scenarios where the target imagery comprises natural scenes from commercial stock providers, our investigation encompassed the use of both DALL-E 3 and Midjourney as the underlying text-to-image APIs. As delineated in Table 4, DALL-E 3 and Midjourney exhibit comparable performance in terms of image similarity metrics. However, a qualitative assessment reveals that Midjourney's outputs are often more lifelike and convincing. This observation is

| Text-to-Image API | BLIP2 | CLIP Interrogator | Our Attack |
|---|---|---|---|
| DALL-E 3 | 0.7509 | 0.7044 | **0.8284** |
| Midjourney | 0.7516 | 0.7480 | **0.8132** |

Table 4: Image similarity scores for our attack on natural images using different text-to-image APIs, compared with baseline models.

consistent with the platform's design philosophy, which emphasizes artistic expression and photorealism. An illustrative example contrasting the outputs from both APIs, with respect to a natural image from a commercial stock collection, is provided in Figure 6.

## 6 Limitations

Like any other method, our approach may encounter failure cases due to the complexity of its multi-component pipeline. Potential reasons for these failures include:

1. The fine-tuned CLIP model might not retrieve related keywords if the words are rare, or the model itself may fail to identify the closest texts to the images.
2. The multi-label classifier might not select appropriate modifiers.
3. GPT-4V might not accurately extract related keywords, named entities, and modifiers, or it might fail to generate an appropriate prompt.
4. The text-to-image models might not produce images that align well with the text.

The inherent uncertainty in both multimodal LLMs and text-to-image models complicates the task. We include examples from setting 1, which yielded the lowest image similarity scores, to explore the factors leading to suboptimal results. These examples and their analyses are available in Appendix C. The multi-component nature of our attack pipeline presents numerous opportunities for refinement. Each component, from the fine-tuning of the CLIP model and the training regimen of the multi-label classifier to the optimization of modifiers extracted from data, holds the potential to elevate the attack's effectiveness. Moreover, the art of prompting GPT-4V is not monolithic; alternative prompt constructions could yield improved results within our cyclic approach. While this paper establishes a solid foundation, the practical enhancements of each component present as promising directions for future work, offering prospects for even more complex attacks.

## 7 Cost Estimation

### 7.1 Financial Cost

To justify the practicality of our attack, we provide an estimation of its associated costs. These are mainly divided into two main components: the cost of using GPT-4V and the cost associated with the Text-to-Image API. We detail each as follows:

**GPT-4V Cost.** According to OpenAI's pricing, there is a charge per million tokens for both input and output. To estimate the cost for using GPT-4V, we consider the number of input and output tokens required per round of our attack. Based on our analysis, the average number of input tokens is 900 for the initial prompt and 1165 for each subsequent round. The average number of rounds to achieve maximum similarity is approximately 2.7, leading to a total of 4395 input tokens per test sample. It's noteworthy that the examples from natural images shown in Figure 7 demonstrate that even with 1-2 rounds, we can achieve

an approximation of the targeted image. For output tokens, the average is 381 per round. Given the current rates on OpenAI's website, the total cost for using GPT-4V, considering both input and output, is approximately $0.09. Additionally, a small fee of approximately $0.03 applies for including images in queries, based on the average number of rounds.

**Text-to-Image API Cost.** Our experiments utilize either Midjourney or DALL-E 3, so we calculate the cost for each separately. Based on the rates from Midjourney's and OpenAI's websites, the cost per image generation is about $0.03 for Midjourney and $0.04 for DALL-E 3 (for 1024*1024 resolution images). Consequently, the total generation cost using Midjourney is approximately $0.12, and for DALL-E 3, it's around $0.16.

**Overall Cost.** The total cost per sample depends on the text-to-image API used. If the attacker utilizes Midjourney, the cost is approximately $0.235 per sample, and for DALL-E 3, it is around $0.275. These costs are substantially lower than the typical prices in AI-generated image marketplaces (ranging from $3 to $7) and significantly below the cost of stocks of real images (which can be $50 to $500).

## 7.2 Computational Cost

Fine-tuning the CLIP model with 1.37 billion parameters and training the classifier are two critical components of our approach. Fine-tuning the CLIP model for 10 epochs on 2 million samples require the use of four A100 GPUs and takes six days to complete. Additionally, training the Multi-Layer Perceptron (MLP) classifier on approximately 360,000 samples for 100 epochs is executed in just 25 minutes on a single A100 GPU.

# 8 Ablation Study

To highlight the significance of the additional information provided by the CLIP model and the classifier, we conduct an ablation study within one of our experimental settings (setting 1). In this experiment, we use the same GPT-4V prompts as described in Tables 6 and 7, but exclude information from the CLIP model and the classifier. We also include specific tips and instructions to demonstrate how prompt engineering operates in Midjourney, incorporating an example of a prompt-image pair from Midjourney. We assess both variants—with and without the example—achieving CLIP scores of 0.8415 and 0.8395, respectively. For comparison, the CLIP scores for BLIP2, CLIP Interrogator, and our approach in setting 1 are 0.8477, 0.8874, and 0.8946, respectively. These results underscore the critical role of the information extraction component in enhancing the effectiveness of our strategy.

# 9 Conclusion

This research has unveiled a novel attack strategy in the realm of digital imagery, specifically targeting AI-generated image marketplaces and premium stock image providers. By effectively employing state-of-the-art multi-modal models, our method has demonstrated the ability to generate visually comparable content at a significantly reduced cost, challenging the current economic dynamics of the digital imagery landscape. Moreover, the compilation of a large-scale dataset from Midjourney is a pivotal contribution to this field. This dataset not only aids our research but also serves as a valuable resource for future studies, offering insights into the capabilities and challenges of text-to-image technologies. Additionally, our approach can serve as a powerful tool for data augmentation, generating varied visual content to enhance machine learning model training, which is often hampered by the expensive and time-consuming nature of traditional data collection. In conclusion, our study highlights new threats against digital imagery, stressing the need for urgent action in identifying and countering these risks. Future research should aim at developing protective measures to preserve digital image integrity amidst advancing AI technologies.

## Ethical Considerations

This paper presents an attack designed to generate images similar to specific, selected stock images at a fraction of their actual cost. The primary aim of this research is to highlight potential new avenues for misuse of multimodal LLMs against the business models of stock image services and content creators. We believe it is vital to understand such threat models to develop appropriate defensive mechanisms effectively. While watermarking techniques (Wen et al., 2023; Zhao et al., 2023; Zhang et al., 2024) effectively protect content, assessing visual LLMs' resilience against these measures is vital. Therefore, exploring robust countermeasures is essential for future research.

In recognition of the potential risks posed to image creators by such attacks, we take several measures to minimize the possibility of reproduction. We will not freely publicize the code, the fine-tuned CLIP model, or the classifier. Additionally, the data used to fine-tune the CLIP model will be excluded from the Midjourney dataset that we intend to release publicly, further mitigating the risk of misuse.

## Acknowledgements

This work was supported by the NSF grant 2131910 and by the Young Faculty Award program of the Defense Advanced Research Projects Agency (DARPA) under the grant DARPA-RA-21-03-09-YFA9-FP-003.

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

## A  Data Collection

In this section, we discuss our approach to gathering data from Midjourney. First, we provide a brief overview of Midjourney and its functionality. Next, we delve into the specific steps of our data collection process. Finally, we describe the characteristics of the data we collected from Midjourney. This offers readers a clear picture of the foundation of our research.

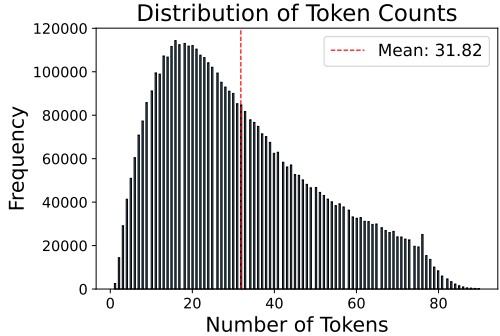

Figure 5: The distribution of the number of tokens in prompts from Midjourney samples.

### A.1  Introduction to Midjourney

Midjourney is a powerful black-box text-to-image API that generates images from natural language descriptions, commonly referred to as "prompts." It operates in a manner analogous to other AI tools like OpenAI's DALL·E and Stability AI's Stable Diffusion. Midjourney has consistently advanced its algorithms, launching new model versions periodically. The algorithm's Version 2 was introduced in April 2022, followed by Version 3 in July 2022. Alpha iterations of Versions 4 and 5 were released in November 2022 and March 2023, respectively. The enhancements in Version 5.1 leaned towards distinct image stylization, whereas its RAW variant optimized interpretations for clearer prompts. The subsequent Version 5.2 further refined the output image quality.

Midjourney operates exclusively via a Discord bot present on their official server. Users can engage with this bot directly or have it on a different server. To produce images, the /imagine command is employed along with a desired prompt, resulting in a set of four generated images. Users then have the option to upscale their preferred images.

## A.2   Data Crawling

The Midjourney server offers a variety of channels designed for collaborative work, including dedicated rooms for beginners, themed generations, and general discussion areas. Both prompts and the corresponding generated images are posted within these channels. We target the general channels, which contain a broader range of generations compared to theme-specific channels. In our approach, we target 10 such general channels. It has to be noted that for our data collection, we utilize DiscordChatExporter,[11] a tool adept at extracting chat histories from Discord channels. This tool facilitates the extraction of chat records from Discord, enabling us to handle direct and group messages, export from various channel types, and retain rich media and specific Discord formatting nuances.

## A.3   Data Characteristics

In our data collection process, we successfully crawled over 20 million samples. These encompass duplicated captions associated with distinct generated images. Among these, more than 12 million samples have unique captions. The collected samples can be categorized into two main groups. The first group consists of samples displaying a grid of 4 generated images. The second group presents samples with a single upscaled image.

The collected raw data include several features: *AuthorID*, *Author*, *Date*, *Content*, *Attachment*, and *Reactions*. Both *AuthorID* and *Author* remain consistent for all samples, pointing to Midjourney's BOT. The *Date* field contains timestamps in the Unix timestamp (or Epoch time) format, which counts the number of seconds that have elapsed since January 1, 1970 (UTC). An example of such a timestamp is 1681516818. *Attachment* denotes the URL of images corresponding to each caption. Lastly, *Reactions* pertains to the emoji reactions associated with the data.

However, the *Content* feature requires a more detailed description. The *Content* comprises the following items:

1. **URL (Optional)**: Users may embed the URL of an image in their prompt, particularly if they wish to modify or obtain different variations of an image.
2. **Main Prompt**: The primary textual prompt provided by the user.
3. **User's ID**: The identifier of the user who submitted the prompt.
4. **Type**: Specifies whether the image is a high-resolution upscaled version or a grid containing lower resolution variations.
5. **Parameters**: Parameters are options added to a prompt that change how an image is generated. They can alter aspects such as the image's aspect ratios, switch between Midjourney model versions, and much more. Parameters are always appended to the end of a prompt, following a double dash ('–'). Multiple parameters can be included in each prompt. Detailed documentation of each parameter is available on Midjourney's website.

## A.4   Data Pre-processing

We implement various data processing steps on this expansive dataset. Some key steps include:

1. Extracting the original prompt from the *Content* feature.
2. Removing empty rows or rows with NaN values.

---

[11]https://github.com/Tyrrrz/DiscordChatExporter

3. Categorizing samples based on whether their prompts contain an image URL. For our method, we exclude these samples, ensuring none are included in our training data.

4. Converting the Unix timestamp in the *Date* feature into a more standard date format.

5. Determining the version of the model using the *Date* feature and the known release dates of each version.

6. Excluding rows where prompts contain emojis or non-English words.

Lastly, from our processed dataset, we select approximately 4.8 million samples with upscaled images. We particularly choose prompts with 77 tokens or fewer, considering the constraint of the CLIP model which only accepts text inputs of up to 77 tokens. The distribution of token counts in these prompts is illustrated in Figure 5.

### A.5 Collecting Modifiers

To extract modifiers from our large-scale dataset, we analyze the common format of Midjourney's prompts. Upon reviewing numerous samples, we observe that prompts often contain a general description followed by modifiers, typically separated by commas. While this pattern is not always consistent, it is prevalent enough in a large dataset to yield a significant set of frequent modifiers. After examining millions of samples, we identify approximately 1800 modifiers, each occurring more than 1000 times in the dataset. Table 5 presents some of the most frequent modifiers along with their corresponding frequencies.

## B Human Evaluation

For the human evaluation component of our study, we adopt a methodology similar to that done by Li et al. (2022b). As outlined in the experimental section, we randomly select five samples from each of the seven settings, amounting to a total of 35 samples. These are then presented to five annotators, who are instructed to score each image using a 5-point Likert scale. The user interface for this task is a Google Form consisting of 35 multiple-choice grid questions, one for each sample.

| Modifier | Frequency |
|---|---|
| 8k | 0.0882 |
| octane render | 0.0453 |
| photorealistic | 0.0442 |
| cinematic | 0.044 |
| realistic | 0.0329 |
| 4k | 0.0324 |
| highly detailed | 0.0297 |
| cinematic lighting | 0.0266 |
| hyper realistic | 0.0257 |
| unreal engine | 0.0249 |

Table 5: The 10 most frequent modifiers extracted from the Midjourney dataset, with frequencies indicating the percentage of prompts containing each modifier relative to the total number of samples.

In each question, annotators are presented with four images: the original image at the top, and three other images below it, generated by our attack and the baselines, shuffled in no specific order. The task requires the annotators to assign a similarity score ranging from 1 (not similar at all) to 5 (very similar) for each of the three shuffled images, based on its resemblance to the original image. Detailed instructions and scoring guidelines are provided to the annotators to ensure a consistent and accurate evaluation process. Here are the instructions and scoring guide:

**Instructions to Annotators:**

"In this task, you will be shown a set of images. For each set, the top image is the original, which may be a grid of four images showing different variants of a single generation. Below the original are three generated images. Your role is to assess how similar each of the three generated images is to the original image (or its variants) and give a similarity score from 1 to 5. Score 1 for 'Completely Different' if there are no discernible similarities, and score 5 for 'Very Similar' if the generated image nearly identically matches

the original in content, style, and details. Use the intermediate scores to indicate varying degrees of similarity."

**Scoring Guide:**

1. **Completely Different**: No discernible similarities exist between the generated image and the original, presenting entirely distinct content and style.

2. **Barely Similar**: The original and generated images may share some elements or themes, but they significantly differ in both content and style.

3. **Somewhat Similar**: The generated image bears a resemblance to the original in terms of content or subject matter, despite noticeable differences in style.

4. **Closely Similar**: Minor variations are present, but the generated image generally mirrors the original in content and subject matter.

5. **Very Similar**: The generated image closely resembles the original in content, style, and fine details, indicating a strong likeness.

## C   Failure Cases

In Figure 8, we present three examples from setting 1 that receive the lowest image similarity scores, highlighting specific instances where our method does not perform as expected.

In the first failure case, our method successfully identifies several elements of the image, including the Asian woman, the expression, and the water pouring over her. However, the images are not deemed highly similar. A potential reason for this discrepancy is the word "Awkwafina". Although our named entity extractor feeds this keyword alongside other related keywords to GPT-4V, the model fails to select it as a related element of the prompt, leading to a less accurate representation in the generated image.

The second failure case also exhibits a mismatch. While our method extracts many elements and features from the image, the degree of similarity is not substantial. The original prompt contains special names that our method does not extract. Furthermore, Midjourney fails to accurately render images of these characters, instead generating unrelated figures. This outcome underscores a dual failure: both our named entity extractor and the text-to-image model do not fulfill their intended functions effectively.

In the final example, the simplicity of the original prompt grants Midjourney significant freedom in varying most aspects of the image, complicating our method's ability to generate a singular prompt that accurately represents all images simultaneously. This issue arises because we use a completely random sample from Midjourney's outputs to construct our test dataset, resulting in some instances with less effective prompts.

These cases underline the complex challenges in prompt-based image generation, where both the extraction of relevant information and the generative capabilities of the model play critical roles in achieving accurate results.

| Text-to-Image API | Original Image | BLIP2 | CLIP Interrogator | Our Attack |
|---|---|---|---|---|
| DALL-E 3 | | | | |
| Midjourney | | | | |
| DALL-E 3 | | | | |
| Midjourney | | | | |
| DALL-E 3 | | | | |
| Midjourney | | | | |

Figure 6: Comparison of natural images from two different Text-to-Image APIs with those generated by our method, BLIP2, and CLIP Interrogator.

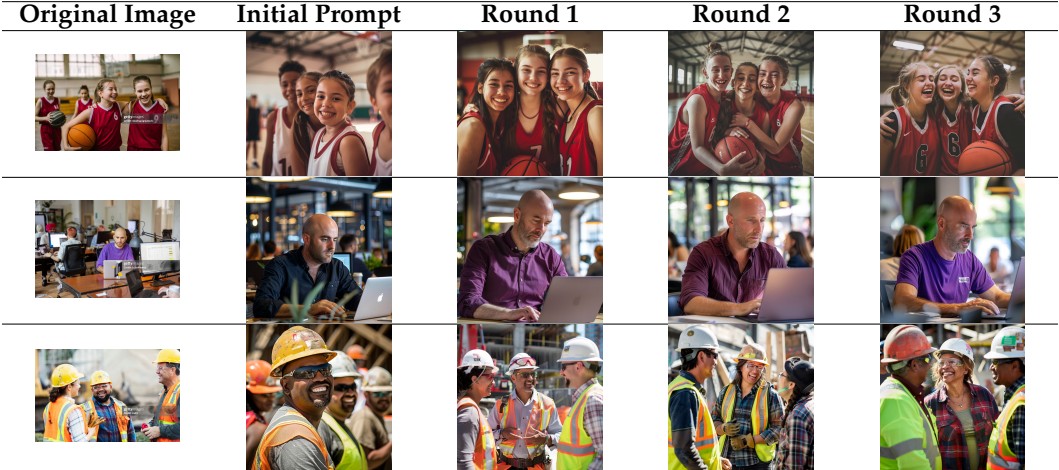

| Original Image | Initial Prompt | Round 1 | Round 2 | Round 3 |
|---|---|---|---|---|

Figure 7: Examples of natural images where our attack method achieves an acceptable approximation of the original model with only 1-2 rounds of refinement.

| Original Prompt | Original Image | Generated Prompt | Generated Image |
|---|---|---|---|
| Gallons of water are pouring down on Awkwafina's head, soaking wet, full body –v 5 –q 2 |  | An 8k high-definition, realistic, detailed photo-rendered portrait of Constance Wu from the waist up, caught off guard and visibly reacting to being suddenly soaked in a torrential downpour during daytime ... |  |
| A group photo of Alex Silva, Jesse Chevan, Justin Gonçalves, and Charlie Hack; 30 years old, portrait, editorial photograph, 8k, weird and odd –s 750 –v 5 |  | Early 1970s Kodak Portrait style portrait of a group of nerdy young men with curly hair and vintage round glasses, dressed in muted, natural-colored academic attire, photographed against a simple backdrop with even, ... |  |
| adult kids with black hairs talking real image |  | Premium 8k resolution portrait conveying a nuanced, refined family dynamic, with subtle punk fashion influences and minimal tattoos. The atmosphere should be moody yet sophisticated, with attire that is stylish and understated, ... |  |

Figure 8: Three examples of failed cases with the lowest image similarity from setting 1, showcasing the original and generated prompts alongside their corresponding images.

---

**Prompt template to start the cycle**

---

**Task**: You are tasked with analyzing a grid of four images generated by a text-to-image model, keeping in mind that they represent varied interpretations from a single prompt, rather than a series of similar objects or scenes. The objective is to deduce the original prompt used to create these images. Your analysis should focus on identifying common themes, elements, or characteristics present across all four images. The challenge is to construct a descriptive prompt that encapsulates these observations. Here are some instructions to generate the prompt:

- Be within 15-50 tokens in length, as per the CLIP tokenizer.

- Encompass all common features visible in the images.

- Examine the provided list of modifiers. Select and include those you deem relevant and which meaningfully contribute to capturing the essence of the images: ...list of modifiers here...

- Prioritize the integration of names from the provided list into your analysis of the image grid, especially when they align with the context or themes depicted. When creating the prompt, if you recognize a famous individual in the image and their name is included in the special names list I've provided, please use that name directly in the prompt. Avoid using phrases like 'a character resembling someone' and instead use the actual name from the list, provided it aligns with the person identified in the image: ...list of named entities here...

- Choose general words from the list that are relevant to the images, reflecting their themes and atmosphere. After ensuring relevance, use these words to enhance the prompt's descriptiveness: ...list of general words here...

- Avoid phrases like 'Create a series of' or using plural terms that might lead the image generation API to produce multiple versions in each variation.

- Typically, follow a structure starting with a general description, followed by specific features and modifiers, separated by commas: *"Dark street Tokyo environment Traditional Japanese illustration of a Funky Musician. With bird mask traditional Japanese elements and neo electro string instrument big japanese graffiti, Crisp contemporary illustrations, bold lines, vibrant, metallic, moving, geometric, expressive"*

---

Table 6: The initial prompt template provided to GPT-4V to begin the iterative refinement cycle. The sections highlighted in red represent the information obtained from the fine-tuned CLIP model and the multi-label classifier.

---

**Prompt template for refining the prompts in the cycle**

**Task:** You are tasked with analyzing a grid of four images generated by a text-to-image model (The first grid). Keeping in mind that they represent varied interpretations from a single prompt, rather than a series of similar objects or scenes. The objective is to deduce the original prompt used to create these images. Your analysis should focus on identifying common themes, elements, or characteristics present across all four images. The challenge is to construct a descriptive prompt that encapsulates these observations.

**Instructions:** Here are some instructions to generate the prompt:

- Be within 15-50 tokens in length, as per the CLIP tokenizer.

- Encompass all common features visible in the images.

- Examine the provided list of modifiers. Select and include those you deem relevant and which meaningfully contribute to capturing the essence of the images: ...list of modifiers here...

- Prioritize the integration of names from the provided list into your analysis of the image grid, especially when they align with the context or themes depicted. When creating the prompt, if you recognize a famous individual in the image and their name is included in the special names list I've provided, please use that name directly in the prompt. Avoid using phrases like 'a character resembling someone' and instead use the actual name from the list, provided it aligns with the person identified in the image: ...list of named entities here...

- Choose general words from the list that are relevant to the images, reflecting their themes and atmosphere. After ensuring relevance, use these words to enhance the prompt's descriptiveness: ...list of general words here...

- Avoid phrases like 'Create a series of' or using plural terms that might lead the image generation API to produce multiple versions in each variation.

- Typically, follow a structure starting with a general description, followed by specific features and modifiers, separated by commas: *"Dark street Tokyo environment Traditional Japanese illustration of a Funky Musician. With bird mask traditional Japanese elements and neo electro string instrument big japanese graffiti, Crisp contemporary illustrations, bold lines, vibrant, metallic, moving, geometric, expressive"*

Your generated prompt given the information above: ...generated prompt from the previous round...

Consider the second grid of images that were generated using the prompt I obtained from you above. Compare these images with the original image I first provided (the first grid of images). Your task now is to refine your prompt to achieve a closer resemblance to the original image. Identify areas where the generated images differ from the original, focusing on style, subject matter, themes, and other key elements. Utilize all provided information above which are related to the images, including different artistic styles, names of entities, or other relevant elements, to make the necessary adjustments. The objective is to modify your previous prompt in such a way that it results in images more accurately reflecting the original in appearance, atmosphere, and concept. Suggest specific changes, whether it involves a different artistic style, more accurate incorporation of certain entities or names, or thematic adjustments. The goal is for the improved prompt to bridge the gap between the generated images and the original image.

---

Table 7: The prompt provided to GPT-4V to refine the prompt at each round based on the generated image in the previous round.

