# OpenReview forum: "Iteratively Prompting Multimodal LLMs to Reproduce Natural and AI-Generated Images"
_colmweb.org/COLM/2024/Conference — COLM_

### Official Review · Reviewer_1ZQM · 2024-05-10

**Rating:** 6
**Confidence:** 4
**Ethics Flag:** 2

**Summary:**

This paper proposed a method to recover the prompt used to generate the input image and use the prompt the reproduce the image. The method is basically a captioning system consisting of (1) a fine-tuned CLIP model for constructing a pool of image and text embeddings, which will be used to retrieve relevant information for the input image, (2) a multilabel classifier to predict the modifiers given the image, and (3) use GPT-4V to generate and refine the final prompt given the information from (1) and (2) with the image to generate the prompt. The proposed method performs better than two off-the-shelf captioning models.

**Ethics Concerns Details:**

The method in this paper can be used to plagiarize copy-righted artworks. The authors should discuss the potential impact of using this method.

**Reasons To Accept:**

The method is technically sound and could have broad applications.

**Reasons To Reject:**

1. **Unfair baseline comparison.**

This is the main problem for this paper, and important baselines are missing for a fair comparison to evaluate the effectiveness of the proposed method. This paper only chooses two off-the-shelf captioning models as baselines, while the proposed model is trained on **additional data** (2M samples for fine-tuning CLIP and 300k examples for modifier classification). The **model size** is also not comparable. The proposed model uses the backbone ViT- G/14 pretrained on LAION-2b. Do the BLIP2 and CLIP Interrogator use the same backbone and pretrained on a similar amount of data? Moreover, the proposed system uses GPT-4V, which is a big advantage.

To make a fair comparison, the following baselines are suggested:
1) fine-tune the baselines with the same amount of data as the proposed method uses.
2) make sure the model size is comparable.
3) use GPT-4V as the captioning model.

2. **Lack of ablation studies**

This is also important to evaluate the effectiveness of the model design. The proposed pipeline involves multiple steps: fine-tuning, multi-label classifier, and GPT-4V to summarize based on the entities, keywords, etc. Are all those steps necessary? Which step is more important to the final performance? This version of the paper did not carefully ablate those components. The performance gain over the baselines could be just because of using more data and a bigger model, as mentioned before.

Overall, this paper has problems in baseline comparison and lacks ablations, making it difficult to evaluate the method's effectiveness.

---

> ### Author Rebuttal · Authors · 2024-05-31
>
> We appreciate the detailed review and thank you for highlighting these concerns.
>
> The reviewer expressed concerns that this paper only chooses two off-the-shelf captioning models as baselines. To the best of our knowledge, there are two suitable open-source tools for this task: CLIP Interrogator and captioning models, for which we selected BLIP2 due to its recent advancements and effectiveness in image captioning. Besides, there are no existing baselines with large multimodal LLMs.
>
> The reviewer suggested fine-tuning the baselines on the same dataset we used. However, this approach presents some difficulties. The CLIP Interrogator, which merges CLIP with BLIP for optimized text prompts, complicates practical fine-tuning on new datasets due to its integrated nature. Moreover, the smallest BLIP2 version holds 3.87 billion parameters, significantly larger than the 1.37 billion parameters of our CLIP model. Our prior experience required 2 million samples and six days on four A100 GPUs to fine-tune the CLIP model, indicating that fine-tuning BLIP2 is not feasible with our resources.
>
> The reviewer expressed some concern regarding the comparability of the sizes of the baselines and our model. It is difficult to find models with exactly matching sizes for specific tasks. However, the parameter counts of our models and the baselines are generally comparable. The baseline BLIP2 model contains 3.87 billion parameters, whereas the CLIP Interrogator, combining CLIP and BLIP, totals about 898 million parameters. The specific CLIP model used in our approach has 1.37 billion parameters.
>
> The reviewer asked for an ablation study that shows the importance of each component. We conducted an experiment by removing the information from the CLIP model and the classifier in one of our experimental settings (Setting 1). We used the same GPT4-V prompt, excluding these components' data. Additionally, we incorporated specific tips and instructions to demonstrate how prompt engineering operates in Midjourney, including an example of a prompt-image pair from Midjourney. We tested the settings with and without the example, obtaining CLIP scores of 0.8415 and 0.8395, respectively (The CLIP scores for BLIP2, CLIP Interrogator, and our approach are 0.8477, 0.8874, and 0.8946, respectively in setting 1). This comparison underscores the essential role of the information extraction component in our strategy's success. These findings will be detailed in the final paper.

---

> > ### Comment · Reviewer_1ZQM · 2024-06-03
> > **Reply to Rebuttal**
> >
> > Thank you for answering my comments. Please include those discussions and ablations in the paper, which will make it more solid. I raised my score.

---

### Official Review · Reviewer_imxU · 2024-05-10

**Rating:** 8
**Confidence:** 4
**Ethics Flag:** 1

**Summary:**

This paper explores a novel and intriguing question in the realm of text-to-image system safety: Is it possible to recover the prompt used to generate an image using a text-to-image system? Given the potential value of high-quality prompts in the AI-generated content (AIGC) domain, this question is worth investigating.

The paper assumes that the attacker has black-box access to all text-to-image APIs without prior knowledge of the underlying models and data, and that the generated images are accessible. This setup seems to be very reasonable.

The proposed method involves fine-tuning CLIP on a dataset of 4.8 million prompt-image pairs to generate text and image embeddings. Similar images to the target images are retrieved, and the corresponding text in the database is used to extract key entities. A multi-label classifier is trained based on the image embeddings to extract image modifiers from a set of 1,800 high-frequency modifiers. GPT-4V is then employed with task demonstration, instructions, the target image, and extracted information to generate the prompt. This generation process can be refined iteratively to improve accuracy.

Evaluation results demonstrate that the proposed framework is more effective in terms of the quality of generated images, with Figure 4 providing clear illustrations of the cases. Furthermore, the method can be transferred from DALL-E 3 (source) to Midjourney (target), and the overall cost (excluding training) is low, at approximately $0.24 per sample.

**Questions To Authors:**

Minor Suggestions:

- When estimating the cost, it would be beneficial to consider the computational cost for training CLIP and the classifier. Although these costs are one-time and can be amortized later, it would still be informative to know the training cost.
- Regarding the evaluation, adding an oracle analysis by comparing the generated prompts with the real prompts used to create the target images would provide interesting insights.


Typos:

- DALL-E 3 Betker et al. (2023) -> DALL-E 3 (Betker et al., 2023): Use \citep instead of \cite for these citations.
- Figure 2: There is an empty period on the new line.

**Reasons To Accept:**

- Overall this paper is well-written, and the reading is enjoyable.

- I like the question explored which might be valuable for future text2image safety research.

- The framework is well-motivated and effective given the current results and the cases in Figure 4.

**Reasons To Reject:**

I did not find anything significant to reject this paper.

---

> ### Author Rebuttal · Authors · 2024-05-31
>
> Thank you to Reviewer for their insightful review and suggestions.
>
> The reviewer asked if we could estimate the computational cost for training the CLIP model and the classifier. Fine-tuning the CLIP model for 10 epochs on 2 million samples using four A100 GPUs took six days. Additionally, training the MLP classifier on ~300K samples for 100 epochs took only 25 minutes using an A100 GPU. We will include these details in the final version of the paper.
>
> We agree with the reviewer that adding an oracle analysis to compare the generated prompts with the real prompts would provide interesting insights regarding the evaluation. As an initial observation, upon comparing the generated prompts with the original ones, we found that the generated prompts are typically longer and more detailed. It's important to note that different prompts can lead to similar images, so the final prompt might not exactly match the original one. Please note that natural images do not have original prompts. We will include a more detailed analysis in the final version of the paper.
>
> The typos, including the citation format and the issue in Figure 2, will be corrected in the final version.

---

> > ### Comment · Reviewer_imxU · 2024-06-04
> >
> > Thanks for your response and I believe these additional cost/oracle analyses could benefit the final version.

---

### Official Review · Reviewer_w2oy · 2024-05-10

**Rating:** 9
**Confidence:** 4
**Ethics Flag:** 1

**Summary:**

This research work includes design attacks for generating images in a refined matter (iterative) that are significantly cheaper than online digital stock images using text-to-image APIs and LLMs.

Participants are shown 4 images and an original image and are asked to tell how similar each image is to the original image using a Likert scale for similarity. From these 4 images, one is generated by the author’s attack, while the others are the baseline-generated images (CLIP interrogator 2 and BLIP2).

Authors conclude that overall, while both midjourney and DALLE-3 (text-to-image) are high-performing, midjourny produces more realistic photos.

The metrics used in the study is CLIP score alongside human evaluation.




Main contributions:

- A large-scale dataset of images from Midjourney collected from their official Discord channel that can enhance the text-to-image line of research.

- Created an attack to create images that have a high similarity scores to the digital images from Web via 1-2 rounds of refining. GPT-4V and midjourney/DALLE-3 are used for the experiments.

- Their work showcases that attention should be paid towards maintaining the integrity of digital images.

**Reasons To Accept:**

Strengths:

- Great comparison of results both visually and quantitatively.

- Cost analysis of the attack shows that with negligible money, images that are very close to the original digital image can be created.

- Authors have provided enough details that makes this research paper reproducible for the most part.

- Providing a set of failure cases that depicts what are some main flaws with the process. Further, expanding on limitations of their current work sheds light on what could be picked up and improve by future researchers.

**Reasons To Reject:**

I am pro accepting this paper. It is a novel concept and beneficial to the community.

---

> ### Author Rebuttal · Authors · 2024-05-31
>
> Thank you to Reviewer for their positive feedback and strong support of our work.

---

### Official Review · Reviewer_BmpJ · 2024-05-12

**Rating:** 7
**Confidence:** 4
**Ethics Flag:** 2

**Summary:**

The paper introduces an original attack strategy for generating images that look like traditional stock images  or images generated with APIs like DALL-E 3 or Midjourney. The proposed strategy uses a fine-tuned multi-modal CLIP model to identify keywords and named entities representative for an image, and also a multi-label classifier to identify relevant modifiers. Such information is provided as instruction to GPT-4V, which finally produces a prompt that can be used to generate the desired images. The authors also perform a cost analysis and show that the proposed attach strategy can be used to generate images at at fraction of the current cost.

**Ethics Concerns Details:**

Generating images that look like real stock images may come with some ethical implications. While the authors present this as an attack strategy, in the wrong hands, the strategy can be misused. Of course, this is a danger posed by many modern deep learning approaches.

**Questions To Authors:**

The authors mention that the dataset assembled from Midjourney will be released publicly. Is that allowed by Midjourney? Can that data be shared?

**Reasons To Accept:**

The authors propose a novel attack strategy and show that they can generate images that look like desirable real stock images or images generated by APIs such as DALL-E 3 or Midjourney at a fraction of the cost.

The authors perform both quantitate and qualitative evaluation of the proposed approach.

The authors clearly point out the limitations of the work.

**Reasons To Reject:**

While the proposed approach can be seen as a novel attack strategy, it leverages existing tools in a pretty straightforward way.

The success of the approach depends on the collection of a large set of prompt/image pairs from Midjourney to fine-tune CLIP.

---

> ### Author Rebuttal · Authors · 2024-05-31
>
> Thank you to Reviewer for their thorough and insightful review.
>
> The reviewer expressed concern that our approach leverages existing tools in a straightforward way. We agree with the reviewer and wish to further emphasize the potential misuse of state-of-the-art APIs to target AI-generated image marketplaces and natural image stocks. Our goal is to highlight these emerging threats and provide a foundational approach that future work can build upon to develop more complex strategies.
>
> The reviewer inquired whether it is allowed under Midjourney's policies to publicly release and share our dataset. We commit to publicly releasing our dataset, as there is nothing in the Midjourney Terms of Service (https://docs.midjourney.com/docs/terms-of-service) that prohibits this action. In fact, these terms explicitly state that anyone is free to use and "remix" images and prompts that are posted on their public Discord.

---

> > ### Comment · Reviewer_BmpJ · 2024-06-04
> >
> > Thank you for addressing my concerns. I am supporting the acceptance of this paper.

---

> > ### Comment · Reviewer_w2oy · 2024-06-07
> > **Reply to author's rebuttal**
> >
> > I read other reviewers' review and authors' answer to them and for the most part it seems we are in tune for accepting the paper however, I agree with reviewer 1ZQM that this discussion needs to be added to the paper.

---

### Decision · Program_Chairs · 2024-07-10

**Decision:**

Accept

**Comment:**

All reviewers appreciated the contribution and especially mentioned its cost efficiency. One reviewer mentions it has "a novel concept and beneficial to the community." Rebuttal period was lively and productive. Based on this, I am recommending acceptance.

For the next version of your paper: I would recommend the authors include the cost analysis mentioned by imxU. We would also like to see the ablations etc. recommended by 1ZQM incorporated into the paper.

On the ethics review: it is expected that the authors will take into account this reviewer's suggestions for minor reframe, and the suggested update to limitations.

[ethics comments from PCs] This paper has been flagged by reviewers for potential concern that the attack can be used to obtain trademark images for free/cheaply and wider discussion has to be made. The motivation of this work can be also be presented in a positive light (e.g., improved image search but by generation) rather than framing it as an attack on stock images. **We strongly recommend addressing these issues and adding/revising an ethics statement as needed.**